# *Enterococcus hirae* Mitral Valve Infectious Endocarditis: A Case Report and Review of the Literature

**DOI:** 10.3390/antibiotics12081232

**Published:** 2023-07-25

**Authors:** Roberta Gaudiano, Marcello Trizzino, Salvatore Torre, Roberta Virruso, Fabio Fiorino, Vincenzo Argano, Antonio Cascio

**Affiliations:** 1Department of Health Promotion, Mother and Child Care, Internal Medicine and Medical Specialties “G D’Alessandro”, University of Palermo, 90127 Palermo, Italy; roberta.gaudiano@community.unipa.it (R.G.); fabio.fiorino@unipa.it (F.F.); 2Infectious and Tropical Disease Unit and Sicilian Regional Reference Center for the Fight against AIDS, AOU Policlinico “P. Giaccone”, 90127 Palermo, Italy; marcello.trizzino@policlinico.pa.it; 3Antimicrobial Stewardship Team, AOU Policlinico “P. Giaccone”, 90127 Palermo, Italy; 4Department of Cardiac Surgery, University Hospital “Policlinico”, 90127 Palermo, Italy; salvatore.torre@policlinico.pa.it (S.T.); vincenzo.argano@policlinico.pa.it (V.A.); 5UOC of Microbiology, Virology and Parassitology, AOU Policlinico, 90127 Palermo, Italy; roberta.virruso@policlinico.pa.it

**Keywords:** *Enterococcus hirae*, infective endocarditis, mitral valve, MALDI-TOF MS

## Abstract

*Enterococcus hirae* is a rare pathogen in human infections, although its incidence may be underestimated due to its difficult isolation. We describe the first known case of *E. hirae* infective endocarditis (IE), which involves the mitral valve alone, and the seventh *E. hirae* IE worldwide. Case presentation: a 62-year-old male was admitted to our department with a five-month history of intermittent fever without responding to antibiotic treatment. His medical history included mitral valve prolapse, recent pleurisy, and lumbar epidural steroid injections due to lumbar degenerative disc disease. Pre-admission transesophageal echocardiography (TEE) showed mitral valve vegetation, and *Enterococcus faecium* was isolated on blood cultures by MALDI-TOF VITEK MS. During hospitalization, intravenous (IV) therapy with ampicillin and ceftriaxone was initiated, and *E. hirae* was identified by MALDI-TOF Bruker Biotyper on three blood culture sets. A second TEE revealed mitral valve regurgitation, which worsened due to infection progression. The patient underwent mitral valve replacement with a bioprosthetic valve and had an uncomplicated postoperative course; he was discharged after six weeks of IV ampicillin and ceftriaxone treatment.

## 1. Introduction

*Enterococcus hirae* is a known pathogen of some animal species, causing endocarditis in chickens, diarrhea in rats, mastitis in cattle, intra-abdominal infection in cats, and sepsis in birds [1]. The first documented case of a human infection dates back to 1988 and was reported by Gilad et al., who described the case of a 49-year-old patient with end-stage renal disease affected by *E. hirae* septicemia [2]. Since then, only a few human cases have been documented in the literature, such as urinary tract infections, infective endocarditis, bacteraemia, and biliary tract infections [1].

Although *Enterococci* are a frequent cause of infection in humans, only 0.4% to 3.03% are reported to be from *E. hirae* species; nevertheless, these data may be underestimated due to its difficult isolation, and its identification could increase using MALDI-TOF: a diagnostic system with a higher sensitivity level compared to previous ones [1,3].

We describe a case of mitral valve infective endocarditis (IE) in a 62-year-old male reviewing the previously reported cases of *E. hirae* IE.

## 2. Case Report

A 62-year-old male was admitted to our department for a five-month history of intermittent fever without responding to antibiotic treatment alongside weight loss, sweating, coughing, and fatigue.

His medical history included mitral valve prolapse causing severe regurgitation, recent pleurisy, and lumbar epidural steroid injections due to lumbar degenerative disc disease.

Twelve days before admission, a TEE performed for persistent fever showed one vegetation on the mitral valve posterior leaflet measuring 4 × 6 mm, and *Enterococcus faecium* was identified by MALDI-TOF VITEK MS on blood cultures. According to the 2023 Duke-International Society for Cardiovascular Infectious Diseases IE Criteria, with two major clinical criteria and two minor clinical criteria, the patient was diagnosed with infective endocarditis and was, therefore, admitted to our Department [4].

At admission, the patient was conscious with a body temperature of 38.5 °C, a pulse rate of 112/min, a respiratory rate of 20/min, oxygen saturation of 98% on room air, and blood pressure of 119/76 mm Hg. Laboratory investigation revealed a white cell count of 4900/microliters, a C-reactive protein of 37.8 mg/L, and procalcitonin of 0.173 μg/L.

On day 1, blood cultures were taken, and intravenous therapy with ampicillin 2 g q4h and ceftriaxone 2 g q12h was initiated, according to antibiotics sensitivity. A good clinical response was obtained, and the temperature was resolved after 24 h.

On the third day after admission, the three blood culture sets drawn on day 1 yielded positive for *Enterococcus hirae* (using MALDI-TOF Bruker Biotyper) with an antibiotic sensitivity profile (according to EUCAST clinical breakpoints v 13.1) the same as the one obtained on *E. faecium* identified before admission (Table 1). Antibiotic therapy was, therefore, confirmed, obtaining a clinical improvement and no growth on blood cultures collected on day 3.

An abdominal CT scan showed splenomegaly (longitudinal diameter 16.4 cm) and renal fascia thickening. The colonoscopy showed no pathologic findings.

On day 17, a TEE was repeated and showed a dilated left ventricle and mitral posterior leaflet prolapse, determining severe regurgitation. Moreover, two vegetations attached to the posterior mitral leaflet of 5 × 6 mm and 6 × 7 mm, respectively, were identified. Therefore, the patient underwent mitral valve replacement with a bioprosthetic valve after four weeks of antibiotic treatment. Mitral valve leaflets cultures and blood cultures taken after surgery were negative.

The patient had an uncomplicated postoperative course and was discharged after a six-week IV ampicillin and ceftriaxone course. The timeline of clinical events is depicted in Figure 1.

## 3. Literature Research

Online research without language restrictions was conducted using PubMed and SCOPUS. This research was performed by combining the terms endocarditis and *Enterococcus hirae* without limits. Furthermore, all references listed were hand-searched for other relevant articles.

The PubMed research identified 19 publications; their meticulous analysis led to 15 eligible articles, and no other articles (not present in PubMed) were found through the SCOPUS research. Overall, six articles describing the history of six patients, published between 2002 and 2020, were further evaluated.

Data regarding clinical characteristics, therapy, and the outcomes of the 6 patients were retrieved alongside that of our present case, which is analytically shown in Table 2. All patients were adults with a mean age of 66.3 years (56–78), and 3 out of 7 were females. Predisposing risk factors were present in 6 out of 7 patients, and none of them had a documented source of infection. Dual antibiotic therapy with two beta-lactams or one beta-lactam with aminoglycosides was administered for at least four weeks in all seven patients, five of which needed replacement valve surgery. All patients survived, although two of them had a relapse after the first treatment.

## 4. Discussion

Infective endocarditis (IE) is a life-threatening disease associated with high mortality and severe complications. *Enterococci* are the third cause of infective endocarditis, after *staphylococci* and *streptococci* [11]. *E. hirae* is a member of the *Enterococcus* genus and is a well-documented cause of infection in animals (e.g., chickens, rats, cattle, cats, and psittacine birds) [1]. Otherwise, infection in humans is uncommon and could be secondary to contact with infected animals or the ingestion of contaminated food [12].

Although *E. hirae* is an important cause of endocarditis in broilers [12], our case is the seventh *E. hirae* IE in humans documented worldwide. Moreover, among the six previously described cases, five involved the aortic valve only, and one involving both the aortic and mitral valve. This is the first documented case of *E. hirae* IE involving only the mitral valve.

In the present case, the patient was affected by mitral valve prolapse with severe valve regurgitation. Predisposing risk factors for cardiac pathology were also present in five out of six previous patients, such as a history of coronary artery disease with percutaneous coronary intervention, aortic valve replacement with a bioprosthetic valve, cardiac arrhythmia with surgical ablation and patent foramen ovale, severe aortic regurgitation, severe aortic stenosis and the bicuspid aortic valve [5,6,7,8,9].

Although *E. hirae* is a common pathogen in animals, it is not documented as a clear *E. hirae* exposure in the previously reported cases nor in our one before symptom onset [5,6,7,8,9,10].

Four out of six previous cases of *E. hirae* IE had a history of gastrointestinal disease (colonic polyps with adenoma [6], gastric leiomyoma removal [7], cholecystectomy [9], colorectal cancer [10]); therefore the digestive tract may be the source of infection, as suggested by Winther M et al. [10]. In the present case, the patient had no GI symptoms or history of gastrointestinal disease, and the colonoscopy ruled out any local pathological involvement. The source of entry could be the lumbar epidural steroid injections the patient received a few months before the onset of symptoms due to lumbar degenerative disc disease or the ingestion of contaminated food.

The first blood culture taken before admission to our department identified *E. faecium*, whereas *E. hirae* was correctly identified to the species level through the three blood cultures drawn on day 1 of hospitalization in our department. In both cases, species identification was performed by mass spectrometry analysis MALDI-TOF MS (respectively, by VITEK MS and Bruker Biotyper).

Difficulties in the identification of *E. hirae* were also encountered in the first two documented cases of *E. hirae* IE, in which a rapid ID 32 Strep system misidentified the genus as *E. durans* and *E. gallinarum*. In both cases, *E. hirae* was subsequently and correctly identified using *sodA*_int_ gene sequencing [5,6]. A case of native-valve bacterial endocarditis caused by *Lactococcus garvieae*, initially identified as *E. hirae* IE, was described as well [13].

In the other known *E. hirae* IE cases, species identification was performed by MALDI-TOF MS in cases four, five, and six, while it was unknown in the third.

MALDI-TOF MS proved to be an accurate system for the identification of both usual and unusual *Enterococci* with a high level of species-level discriminatory power [3]. The low rate of *E. hirae* incidence may have been the consequence of the limited power of other diagnostic systems; indeed, the identification of *E. hirae* increased after the development of MALDI-TOF MS [14]. Both the Bruker Biotyper and VITEK MS systems are able to correctly identify a wide range of bacteria; however, VITEK MS has a higher error rate compared to Bruker Biotyper in the case of unusual microorganisms [15], as corroborated by our case in which VITEK MS inaccurately identified the microorganism grown in the blood culture as *E. faecium*.

All seven patients survived *E. hirae* IE, five of which needed valve replacement surgery. The first two patients relapsed after treatment with ampicillin or amoxicillin and gentamicin, followed by rifampicin for six weeks. In the first case, vancomycin and gentamicin were administered for six weeks after relapse; in the second, the initial therapy was successfully repeated [5,6]. The third patient was treated with rifampin and ampicillin, followed by amoxicillin for a total of six weeks [7]. Ceftriaxone was administered in the fourth and fifth cases for six weeks, in combination with ampicillin in one case and with ampicillin, followed by penicillin G and then by a chronic suppressive therapy of oral penicillin in the other [8,9]. The sixth patient received benzylpenicillin for four weeks associated with gentamicin in the first two weeks [10]. In our case, six-weeks of therapy with ampicillin and ceftriaxone was administrated.

The European Society of Cardiology guidelines recommend a prolonged administration (up to 6 weeks) of synergistic bactericidal combinations of two cell wall inhibitors (ampicillin or penicillin and ceftriaxone for penicillin-susceptible strains) or one β-lactams with aminoglycosides for the management of enterococcal IE [16].

The susceptibility of *E. hirae* to antimicrobial agents suggests that this species is more similar to *E. faecalis* than *E. faecium* [14]. Treatment with ampicillin and ceftriaxone, with their synergism by inhibiting complementary PBPs, is as effective as ampicillin and gentamicin for *E. faecalis* IE [17,18]. Therefore, ampicillin plus ceftriaxone seems to be a good therapeutic alternative in *E. hirae* IE, reducing the risk of renal failure, which is typically associated with gentamicin administration [18].

In summary, *E. hirae* is a potential pathogen in severe human infections such as endocarditis; its incidence is expected to increase thanks to MALDI-TOF MS, an accurate system with a higher species-level discriminatory power than previous ones. Treatment with ampicillin and ceftriaxone for enterococcal IE is associated with a good prognosis and a low risk of side effects and could be a good therapeutic choice in *E. hirae* IE.

## Figures and Tables

**Figure 1 antibiotics-12-01232-f001:**
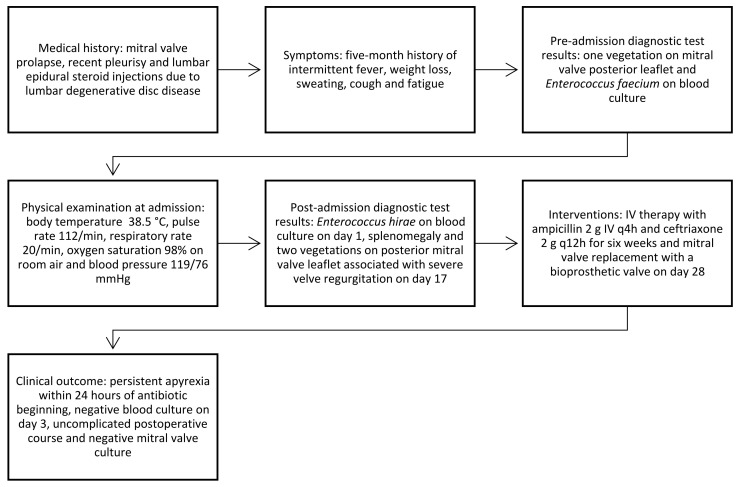
Timeline of clinical events.

**Table 1 antibiotics-12-01232-t001:** *E. faecium* and *E. hirae* antibiograms with M.I.C. calculated by Vitek2.

*E. faecium*	Sensitivity	M.I.C.	*E. hirae*	Sensitivity	M.I.C.
Ampicillin	S	<=2	Ampicillin	S	<=1
Linezolid	S	2	Linezolid	S	1
Teicoplanin	S	<=0.5	Teicoplanin	S	<=0.5
Tigecycline	S	<=0.12			
Vancomycin	S	<=0.5	Vancomycin	S	<=0.5
			Gentamicin (SYN)	S	<=500

**Table 2 antibiotics-12-01232-t002:** Clinical characteristic of patient with *E. hirae* IE.

	Age	Sex	Medical History	Valve Involvement	Predisposing Risk Factors	Identification	Treatment	Relapse
Poyart et al., 2002 [5]	72	M	1-month history of fever, chills, progressive malaise, generalized weakness	Vegetations involving the left and right aortic-valve leaflets	Aortic insufficiency. History of coronary artery disease and percutaneous transluminal coronary angioplasty	sodAint gene sequencing	Ampicillin plus gentamicin followed by rifampin for a total of six weeks.	Yes, treated with vancomycin followed by amoxicillin plus gentamicin for a total of eight weeks; aortic valve replacement with a homograft
Talarmin et al., 2011 [6]	78	F	5-month history of fever, generalized weakness, and a 7 kg weight loss	Normal TEE during the first presentation, vegetation on aortic prosthetic valve at relapse	Aortic valve replacement with a bioprosthetic valve	sodAint gene sequencing	Amoxicillin and gentamicin followed by rifampin for a total of six weeks	Yes, treated with amoxicillin and gentamicin followed by rifampin for a total of six weeks
Anghinah et al., 2013 [7]	56	M	Dysarthria, dysphagia, and left hemiparesis are associated with recent weight loss, fatigue, depressive symptoms, and evening fever	Vegetations involving aortic and mitral valves	Cardiac arrhythmia with surgical ablation and patent foramen ovale	Unknown	Rifampin plus ampicillin followed by amoxicillin for a total of six weeks; aortic valve replacement by a biological valve, plastic of the mitral valve, and correction of the foramen ovale.	No
Ebeling et al., 2019 [8]	70	M	3- month history of bilateral leg edema, dyspnea on exertion, fatigue, mild weight loss	Vegetation on aortic valve, purulent aortic root abscesses	Severe aortic regurgitation, moderate anterolateral hypokinesis	MALDI-TOF	Ampicillin followed by penicillin G plus ceftriaxone for a total of six weeks, then indefinite chronic suppressive therapy with oral penicillin; aortic valve replacement	No
Pinkes et al., 2019 [9]	64	F	Fever, hypotension, 2-week history of lightheadedness	Multiple small vegetations on aortic valve, root abscess beneath the right coronary cusp	Severe aortic stenosis, bicuspid aortic valve	MALDI-TOF	Ampicillin and ceftriaxone for six weeks; aortic valve replacement with a bovine pericardial valve	No
Winther et al., 2020 [10]	62	F	2-week history of diarrhea, vomiting, a 10 kg weight loss, headache, dizziness	Vegetation on aortic valve	No	MALDI-TOF	Benzylpenicillin for 4 weeks plus gentamicin for 2 weeks	No
Present case 2023	62	M	5-month intermittent fever not responding to antibiotics, accompanied by weight loss, sweating, cough, abnormal fatigue	Vegetation of size 4 mm × 6 mm over the posterior leaflet of the mitral valve	Mitral valve prolapses	MALDI-TOF Bruker Biotyper	Ampicillin and ceftriaxone for 6 weeks; mitral valve replacement with a bioprosthetic valve	No

## Data Availability

Not applicable.

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
