# Peer review of "Enterococcus hirae Mitral Valve Infectious Endocarditis: A Case Report and Review of the Literature"

_antibiotics, 2023, doi:10.3390/antibiotics12081232_

Round 1

Reviewer 1 Report

Enterococcus hirae mitral valve infectious endocarditis: a case 2 report and review of the literature

Authors:  Roberta Gaudiano, Marcello Trizzino, Salvatore Torre, Roberta Virruso, Vincenzo Argano and Antonio Cascio

The purpose of this study was to describe a case of mitral valve infective endocarditis (IE) in a 62-years-old male 40 and to review the previous known case of E. hirae IE.

General comments:

The description is quite clear.

Major comments:

-        Table 1: please indicate MIC values (determined by broth microdilution method (BMD) or by E-tests ). Please indicate the recommended breakpoints used (EUCAST ? version ?) ?

-        Please summarize the paragrahs between lines 132 and 148

-        Have you performed serum assays for ampicillin ?

Minor Comments:

-        Line 47: please “Enterococcus faecium” in italics

-        Table 1: please replace “gentamycin” by “gentamicin”

-        Table 1: please remove date on imipenem from table 1(molecule not recommended)

-        Line 104: please replace “Enterococcus hirae” by “E. hirae

-        Line 136: please “sodA” in italics

Author Response

Dear reviewer,

thank you for your comments. Here the responses to your points.

Point 1: Table 1: please indicate MIC values (determined by broth microdilution method (BMD) or by E-tests ). Please indicate the recommended breakpoints used (EUCAST ? version ?) ?

Response 1: We indicated MIC values and the recommended breakpoints used.

Point 2: Please summarize the paragrahs between lines 132 and 148

Response 2: We have made a small summary of the lines, in our opinion a further summary could remove imporant information. We are available to make further changes in the next revisions.

Point 3: Have you performed serum assays for ampicillin ?

Response 3: No, we didn’t.

Point 4: Line 47: please “Enterococcus faecium” in italics

Response 4: ok, done

Point 5: Table 1: please replace “gentamycin” by “gentamicin”

Response 5:  ok, done

Point 6: Table 1: please remove date on imipenem from table 1(molecule not recommended)

Response 6:  ok, done

Point 7: Line 104: please replace “Enterococcus hirae” by “E. hirae

Response 7:  ok, done

Point 8: Line 136: please “sodA” in italics

Response 8:  ok, done

Reviewer 2 Report

The article describes the case of a 62-year-old patient with E. hirae infective endocarditis, and the authors reviewed the other six previously published cases. The case shows endocarditis caused by a very rare and difficult-to-isolate microorganism. The authors document the need for mitral valve replacement despite antibiotic treatment. The article is of interest due to the rarity of the case.

The authors should clarify two points:

1.- The patient underwent his first transesophageal echocardiography 12 days before admission. What happened in those 12 days? Why wasn't he admitted earlier? Did not the patient receive antibiotic treatment during those days?

2.- In the abstract, the authors say, "We describe the first known case of E. hirae mitral valve 17 infective endocarditis". But this is not exact because one of the previously published cases had aortic and mitral involvement. This should be corrected in the abstract.

Author Response

Dear reviewer,

thank you for your comments. Here the responses to your points.

Point 1: The patient underwent his first transesophageal echocardiography 12 days before admission. What happened in those 12 days? Why wasn't he admitted earlier? Did not the patient receive antibiotic treatment during those days?

Response 1: The patient came to our attention just the day before the admission, in those 12 days the patient didn't receive any antibiotic treatment.

Point 2:  In the abstract, the authors say, "We describe the first known case of E. hirae mitral valve 17 infective endocarditis". But this is not exact because one of the previously published cases had aortic and mitral involvement. This should be corrected in the abstract.

Response 2:  OK, done

Reviewer 3 Report

Dear Author(s),

Thank you for your interesting and novel manuscript describing the first known case of E. hirae mitral valve infective endocarditis (IE) and the seventh E. hirae IE worldwide.

I am really satisfied with the quality of your manuscript, since it is original, novel, and accompanied by the appropriate literature review, which is higly important since without it it would be not appropriate for the publication in the prestigious journal such as Antibiotics journal.

Case is well-presented, as per CARE guidelines.

Introduction can be a little more extensive. I would include few relevant general info on E. hirae if I were you, in order to familiarize the readers more before the case presentation.

Case description is well organized. If I were you I would maybe make a flow diagram (as per CARE guidelines) and include the most relevant clinical presentation characteristics, laboratory data, drug administration, other findings, etc. within its structure. It can also serve as a graphical abstract maybe if well conducted.

What is more, if I were you I would provide a comment on Duke criteria for endocarditis. 

Discussion is appropriate as well as conclusion (which is based on most relevant findings).

All in all, compliments! Only minor modifications are necessary.

Author Response

Dear reviewer,

thank you for your comments. 

We included few general information on E. hirae in the introduction and the Duke criteria for endocarditis in the discussion.  We also added a flow diagram with the most relevant clinical events. 

Round 2

Reviewer 1 Report

- these are still not real MICs, but approximate MICs calculated by vitek2 / thank you for determining real MICs or specify on the review that this is not a real MIC.

-please remove from table 1: quinupristin/dalfopristin and amipillin/sulbactam . This is of no relevance

- other answers are satisfactory

Author Response

Dear Reviewer,

thank you for your comments.

we specified that MIC are calculated by vitek2 and removed the two lines in Table 1.